# Genetic Polymorphisms of the Telomerase Reverse Transcriptase Gene in Relation to Prostate Tumorigenesis, Aggressiveness and Mortality: A Cross-Ancestry Analysis

**DOI:** 10.3390/cancers15092650

**Published:** 2023-05-08

**Authors:** Yongle Zhan, Xiaohao Ruan, Jiacheng Liu, Da Huang, Jingyi Huang, Jinlun Huang, Tsun Tsun Stacia Chun, Ada Tsui-Lin Ng, Yishuo Wu, Gonghong Wei, Haowen Jiang, Danfeng Xu, Rong Na

**Affiliations:** 1Division of Urology, Department of Surgery, LKS Faculty of Medicine, The University of Hong Kong, Hong Kong, China; 2Department of Urology, Ruijin Hospital, Shanghai Jiao Tong University School of Medicine, Shanghai 200025, China; 3Division of Urology, Department of Surgery, Queen Mary Hospital, Hong Kong, China; 4Department of Urology, Huashan Hospital, Fudan University, Shanghai 200040, China; 5Fudan University Shanghai Cancer Center & MOE Key Laboratory of Metabolism and Molecular Medicine and Department of Biochemistry and Molecular Biology of School of Basic Medical Sciences, Shanghai Medical College of Fudan University, Shanghai 200032, China

**Keywords:** prostate cancer, aggressive prostate cancer, prostate cancer death, telomerase reverse transcriptase, single nucleotide polymorphism, cross-ancestry, European, Chinese

## Abstract

**Simple Summary:**

Mutation in telomerase reverse transcriptase (*TERT*) has been reportedly related to risks of prostate cancer (PCa). However, prior genome-wide association studies (GWAS) were limited to inconsistency, small-scale, one-outcome phenotype, or single ancestry. In this study, we used two large-scale population datasets from European and Chinese ancestries to comprehensively estimate the association of *TERT* loci polymorphisms with prostate tumorigenesis and severity. Results of this study showed that (1) over half of the risk variants were located at the intron 2 region in both populations; (2) seven novel loci situated at intron 2, intron 6, intron 9, and intron 12 were first identified to be related to PCa risk; (3) SNPs rs2736100 and rs2853677 were significantly associated with aggressive PCa, whereas rs35812074 was marginally related to PCa death; (4) most identified loci were different between Europeans and Chinese. These findings support the evidence regarding *TERT* polymorphisms in relation to PCa risk and prognosis, and indicate the heterogeneous genetic architectures of PCa susceptibility loci among distinct ancestries.

**Abstract:**

Background: Telomerase reverse transcriptase (*TERT*) has been consistently associated with prostate cancer (PCa) risk. However, few studies have explored the association between *TERT* variants and PCa aggressiveness. Methods: Individual and genetic data were obtained from UK Biobank and a Chinese PCa cohort (Chinese Consortium for Prostate Cancer Genetics). Results: A total of 209,694 Europeans (14,550 PCa cases/195,144 controls) and 8873 Chinese (4438 cases/4435 controls) were involved. Nineteen susceptibility loci with five novel ones (rs144704378, rs35311994, rs34194491, rs144020096, and rs7710703) were detected in Europeans, whereas seven loci with two novel ones (rs7710703 and rs11291391) were discovered in the Chinese cohort. The index SNP for the two ancestries was rs2242652 (odds ratio [OR] = 1.16, 95% confidence interval [CI]:1.12–1.20, *p* = 4.12 × 10^−16^) and rs11291391 (OR = 1.73, 95%CI:1.34–2.25, *p* = 3.04 × 10^−5^), respectively. SNPs rs2736100 (OR = 1.49, 95%CI:1.31–1.71, *p* = 2.91 × 10^−9^) and rs2853677 (OR = 1.74, 95%CI:1.52–1.98, *p* = 3.52 × 10^−16^) were found significantly associated with aggressive PCa, while rs35812074 was marginally related to PCa death (hazard ratio [HR] = 1.61, 95%CI:1.04–2.49, *p* = 0.034). Gene-based analysis showed a significant association of *TERT* with PCa (European: *p* = 3.66 × 10^−15^, Chinese: *p* = 0.043) and PCa severity (*p* = 0.006) but not with PCa death (*p* = 0.171). Conclusion: *TERT* polymorphisms were associated with prostate tumorigenesis and severity, and the genetic architectures of PCa susceptibility loci were heterogeneous among distinct ancestries.

## 1. Introduction

Prostate cancer (PCa) has become the second most common male cancer and the fifth leading cause of male cancer death worldwide based on the estimated number from GLOBOCAN 2020 [1]. The incidence and mortality of PCa have been increasing in past decades. In the US, PCa incidence increases by 3% annually, with an estimated 288,300 new cases and 34,700 deaths expected in 2023 [2]. In European countries, the estimated new cases and deaths were 473,300 and 108,100, respectively, in 2020 [3]. In China, up to 125,646 new cases and 56,239 deaths were reported in 2022 [4]. To date, only a few risk factors for PCa have been established. Among them, genetic susceptibility is one of the most critical risk factors.

Telomerase, a ribonucleoprotein complex to prevent telomere shortening, is essential for maintaining chromosomal integrity and stability during cell division. Telomerase reverse transcriptase (*TERT*), a key determinant of the enzymatic activity of telomerase, is reportedly related to both aging and carcinogenesis [5]. Thus, mutations in the *TERT* regions (5p15.33) in relation to tumorigenesis have long intrigued researchers. Prior genome-wide association studies (GWAS) have found polymorphisms in *TERT* were associated with risks of multiple cancers, such as breast, lung, glioma, bladder, testicular, pancreas, prostate, and skin cancer [6,7]. In prostate cancer, single nucleotide polymorphisms (SNPs), including rs4449583, rs10069690, rs13172201, and rs2736098, were identified to be associated with disease risks in the European population [7], while rs2736100 and rs10069690 were found to be associated in the Chinese population [8]. Results from previous studies were limited to either inconsistency or their small scale. For the validation of PCa-related *TERT* SNPs in multiple ancestries, only a cross-ancestry meta-analysis found that rs7726159 and rs2736098 were significantly associated with PCa risk in at least two ancestries [7]. Furthermore, there is a research gap on the association of *TERT* variants with PCa severity and prognosis.

Given a paucity of investigation regarding the genetic architecture of susceptibility loci of prostate tumorigenesis and prognosis across *TERT* regions in distinct ancestries, we, therefore, utilize two large-scale population datasets from European and Chinese ancestries to validate the association of *TERT* loci polymorphisms with PCa, PCa aggressiveness, and PCa death, and to explore the genetic heterogeneity effect among distinct ethnic populations. The primary objective of this study is to provide comparable and comprehensive evidence on *TERT* genetic variations in prostate cancer risk in two different ethnic populations.

## 2. Materials and Methods

### 2.1. Study Population

The study population in this study comprised patients of European and Chinese ancestries. Data from European participants were obtained from the UK Biobank (UKB), a large-scale biomedical database containing a prospective cohort with a wealth of genetic and phenotypic information. Around 0.5 million individuals between 40 and 69 years old were recruited from 2006 to 2010 across the UK [9]. Participants with PCa were recorded through the cancer registry, hospital electronic system and self-report. Data from Chinese participants were obtained from the Chinese Consortium for Prostate Cancer Genetics (ChinaPCa), an ongoing case-control study in which 5000 pathologically diagnosed PCa patients were recruited from the local hospitals in the south-eastern region in mainland China, whereas the 5000 cancer-free controls were recruited from the local community or physical examination centers [10,11]. All participants were collected with informed consent, and the study was approved by the regional ethics committee.

### 2.2. Outcome Phenotype Ascertainment

In UKB, participants were followed up for prostate cancer (C61) and PCa-specific death using records linkage with the regional system of disease surveillance, chronic disease management, and electronic health records (EHRs) based on diagnostic codes from the International Classification of Diseases, 10th revision (ICD-10). Cancer survival time was ascertained from the date of cancer diagnosis to the date of death, loss to follow-up, or 15 November 2022, whichever came first.

In ChinaPCa, malignant neoplasm of the prostate was ascertained by pathological diagnosis from the local hospital. Aggressive PCa was defined as PCa with Gleason score (GS) ≥ 8 or prostate-specific antigen (PSA) value > 20 ng/mL according to the National Comprehensive Cancer Network (NCCN) guideline.

### 2.3. SNPs Selection and Genotyping

To ensure the probable regulatory regions of the *TERT* were included, we extended the upstream and downstream of the initial gene region to 100 kb. SNPs located at chr5:1,150,000–1,400,000 were selected according to the HapMap database. Germline DNA samples of the two datasets were extracted from blood samples via whole blood genomic DNA extraction kit and were further genotyped by using the UK Biobank Axiom array in the UKB dataset [12] and Illumina Human OmniExpress BeadChips in the ChinaPCa dataset [10].

### 2.4. Phasing, Imputation and Quality Control

Pre-phasing and imputation in UKB genotyping data were performed via the SHAPEIT and IMPUTE4 program using the Haplotype Reference Consortium and the merged UK10K and 1000 Genome Phase III reference panels [12]. For ChinaPCa genotyping data, pre-phasing analysis was conducted by Eagle v2.4, and imputation was performed using the Michigan Imputation Server using the minimac4 imputation algorithm [13]. Although different imputation algorithms were used on two populations, the imputation accuracy of these two algorithms was deemed highly consistent in accordance with previous research [13]. A posterior probability of >0.90 was applied to call genotypes during the course of imputation. Poorly imputed SNPs were further excluded on the basis of (a) genotype call rate less than 95%, (b) minor allele frequency (MAF) below 0.01, and (c) *p*-value for the Hardy–Weinberg Equilibrium (HWE) test amid controls lower than 1.0 × 10^−6^.

### 2.5. Statistical Analysis

The association of each SNP with PCa and aggressive PCa was estimated by odds ratio (OR), 95% CI, and corresponding *p*-value, using logistic regression analysis with adjustment for age based on an additive model. The association of each SNP with PCa death for the UKB sample was estimated by hazard ratio (HR), 95% confidence interval (CI) and corresponding *p*-value, using Cox regression analysis with adjustment for age, family history, and Charlson Comorbidity Index (CCI), based on an additive model. In order to evaluate the combined effect of the significant SNPs on PCa risk, a polygenic risk score (PRS) was additionally calculated by summing the number of each significant risk allele carried (0, 1, or 2) for each individual, with every single variant weighted by its effect size (log [OR] for binary traits). Gene-based analysis was performed based on the remission and percentage improvement GWAS *p*-values using MAGMA v1.10 software [14]. The analysis was conducted according to genetic variants and linkage disequilibrium (LD) in two ethnic reference data sets (1000 Genomes European panel and 1000 Genomes East Asian panel), and then SNPs were assigned to genes using the MAGMA NCBI37.3.gene.loc file with a 10-kb window. The gene-based association was estimated using Z statistics and the corresponding *p*-value. Regional plots were generated from LocusZoom (http://csg.sph.umich.edu/locuszoom/ (accessed on 20 March 2023)). LD heatmap was created using LDBlockShow v1.39 software [15]. All statistical analyses were performed under PLINK v1.90 software. Two-tailed Bonferroni corrected *p* < 7.35 × 10^−4^ (0.05/68) for UKB and *p* < 5.32 × 10^−4^ (0.05/94) for ChinaPCa were considered statistically significant, while the *p*-value between Bonferroni correction and 0.05 was deemed marginally significant on SNP association analysis. A two-tailed *p* < 0.05 was considered significant in gene-based analysis.

## 3. Results

### 3.1. Participant Characteristics

A total of 209,694 Europeans (14,550 PCa cases/195,144 controls) and 8873 Chinese (4438 cases/4435 controls) were involved in this study (Table 1). Briefly, PCa patients were of older age and had a higher proportion of a positive PCa family history in both populations. The median comorbidity index was higher in Europeans with PCa than in controls (median CCI: 2.0 vs. 0.0, *p* < 0.001). In the Chinese population, PCa patients had an elevated PSA value compared with their control counterparts (median PSA value: 21.8 vs. 9.8 ng/mL, *p* < 0.001). 52.1% and 37.6% of the cancer patients had high levels of PSA (>20 ng/mL) and GS (≥8), respectively.

### 3.2. Associations between TERT SNPs and PCa Risk

Nineteen SNPs were identified to be significantly related to PCa risk in European ancestry under an additive effect assumption (Table 2). These SNPs were located in three regions (Figure 1A), spanning a 6.8-kb region from the promoter (rs2736109) to intron 2 (rs74682426), a 1.8-kb region from intron 2 (rs4449583) to intron 3 (rs7726159), and a 0.9-kb region from intron 3 (rs4975538) to intron 4 (rs10054203) of the *TERT*. The index SNPs in these regions were rs7712562 (OR = 1.16, 95%CI: 1.11–1.20, *p* = 1.17 × 10^−12^), rs7725218 (OR = 1.12, 95%CI: 1.09–1.15, *p* = 1.34 × 10^−14^), and rs2242652 (OR = 1.16, 95%CI: 1.12–1.20, *p* = 4.12 × 10^−16^), respectively. Among these SNPs, five loci (rs144704378, rs35311994, rs34194491, rs144020096, and rs7710703) were first discovered to be related to PCa risk in European ancestry. The combined effect of all the marginal-to-significant SNPs showed a two-fold higher risk of PCa among the European population (OR = 1.99, 95%CI: 1.77–2.25, *p* < 0.001).

In terms of Chinese ancestry, seven SNPs were found to be significantly associated with PCa risk in the additive model (Table 3). These SNPs were situated in two regions (Figure 1B), spanning a 12.0-kb region from the promoter (rs7712562) to intron 2 (rs530443350) and a 27.5-kb region from intron 2 (rs530443350) to intron 7 (rs2853687) of the *TERT*. The index SNPs in these two regions were rs11291391 (OR = 1.73, 956% CI: 1.34–2.25, *p* = 3.04 × 10^−5^) and rs530443350 (OR = 3.08, 95%CI: 1.21–7.82, *p* = 0.018), respectively. Among these SNPs, two loci (rs7710703 and rs11291391) were first detected to be associated with PCa risk in Chinese ancestry. The combined effect of all the marginal-to-significant SNPs showed a 16% higher risk of PCa among the Chinese population (OR = 1.16, 95%CI: 1.03–1.22, *p* < 0.001).

Three SNPs were identified to be associated with PCa risk in both ancestries, including rs7712562 (OR = 1.16, 95% CI: 1.11–1.20, *p* = 1.17 × 10^−12^ for Europeans; OR = 1.72, 95% CI: 1.29–2.29, *p* = 2.15 × 10^−4^ for Chinese), rs74682426 (OR = 1.15, 95% CI: 1.11–1.20, *p* = 1.04 × 10^−11^ for Europeans; OR = 1.70, 95% CI: 1.27–2.27, *p* = 3.25 × 10^−4^ for Chinese), and rs7710703 (OR = 1.10, 95% CI: 1.06–1.15, *p* = 4.30 × 10^−6^ for Europeans; OR = 1.72, 95% CI: 1.28–2.31, *p* = 3.08 × 10^−4^ for Chinese), among which, the first SNP was located in the promoter region whereas the latter two were situated in intron 2 region (Figure 2).

### 3.3. Associations between TERT SNPs and Aggressive PCa Risk

Two *TERT* SNPs were found to be significantly associated with aggressive PCa risk among PCa cases sample from Chinese ancestry (Figure 1C), including rs2736100 (OR: 1.49, 95%CI: 1.31–1.71, *p* = 2.91 × 10^−9^) and rs2853677 (OR: 1.74, 95%CI: 1.52–1.98, *p* = 3.52 × 10^−16^). These two SNPs were both located in intron 2 of *TERT*. The combined effect of all the marginal-to-significant SNPs showed a 34% increased risk of aggressive PCa among the Chinese population (OR = 1.34, 95%CI: 1.05–1.70, *p* = 0.017).

### 3.4. Associations between TERT SNPs and PCa Death

One SNP, rs35812074, was identified to be marginally related to PCa death among PCa cases sample from European ancestry (Figure 1D). The risk allele (C) of rs35812074 was associated with a 61% increased risk of PCa death (HR = 1.61, 95%CI: 1.04–2.49, *p* = 0.034) with adjustment for age, family history and comorbidity index (Figure 3).

### 3.5. Gene-Based Analysis

Gene-based analysis using 56 SNPs from European ancestry and 79 SNPs from Chinese ancestry showed that *TERT* was significantly associated with PCa among the two ancestries (Z = 7.78, *p* = 3.66 × 10^−15^ for European; Z = 1.72, *p* = 0.043 for Chinese). This gene was also found to be related to aggressive PCa (Z = 2.54, *p* = 0.006). However, we did not observe a gene-based association between *TERT* and PCa death (Z = 0.95, *p* = 0.171) (Table 4).

## 4. Discussion

The present study including two ethnic population datasets shows that polymorphisms in *TERT* are associated with prostate tumorigenesis, aggressiveness, and PCa death. This study is the first to comprehensively illustrate the genetic architectures of PCa susceptibility loci across the *TERT* region among distinct ancestries. Specifically, we found that (1) over half of the variants significantly associated with PCa risk or aggressiveness were located at the intron 2 region in both populations; (2) seven novel loci situated at intron 2, intron 6, intron 9 and intron 12 were first identified to be related to PCa risk; (3) rs35812074 located in intron 9 was observed marginally associated with PCa death.

*TERT*, located at 5p15.33 with 16 exons and 15 introns, has been reported to harbor several susceptibility loci that could influence prostate carcinogenesis. A fine-mapping study using 22,301 PCa cases and 22,320 controls identified multiple risk loci, including rs2853669, rs2853676, rs7725218, and rs2242652, which were situated at the promoter, intron 2, intron 3, and intron 4 regions, respectively [16]. Another recent fine-mapping study using European ancestry population additionally found that rs2853677, rs11414507, rs7705526, and rs35334674 at the intron 2 region, rs10069690 at intron 4 region, as well as rs35033501 at exon 16 region were significant SNPs related to PCa risk [17]. Several susceptibility loci, such as rs7712562 at the promoter region and rs7726159 at the intron 3 region, were identified by some case-control studies [18,19]. In this study, we confirmed the aforementioned SNPs to be associated with PCa.

To the best of our knowledge, the present study was the first to comprehensively evaluate the effect of polymorphisms across the *TERT* on PCa severity and PCa-specific death. SNP rs2736100, located in intron 2, was found to be associated with significant PCa aggressiveness in our study, consistent with the finding from a prior case-case study involving 1210 Chinese PCa patients [8]. SNP rs10069690, another susceptibility locus discovered by that study, however, was not confirmed in our study. In terms of prostate cancer-specific death, we first found a potential susceptibility locus (rs35812074) (*p* = 0.034). This SNP was reported to be associated with lung cancer by a recent GAWS [20]. However, the finding of rs35812074 in relation to PCa death in our study should be treated with caution because the risk allele of rs35812074 was detected as the major allele, and the minor allele frequency of this SNP was lower than 5%. Although the current imputation method ensures the imputation accuracy for the genotypes, result interpretation remains further validated in other GWASs and functional experiments.

In the present study, we found that most of the risk loci were located in intron regions, particularly intron 2 in both ancestries. Introns, at present, have been paid increasing regard to gene regulation, such as genome organization, transcription regulation, and alternative splicing [21]. The intron 2 of *TERT* has been presumed to harbor a putative regulatory region [22] and thus is likely to play a significant role in biological functions. As reported by prior functional studies involving polymorphisms in the intron 2 region of *TERT*, rs2853677 was found located within the Snail1 binding site in a *TERT* enhancer, the risk allele of which disrupted the Snail1 binding site, and the resultant derepressed *TERT* expression increased cancer susceptibility [23]. Confirmed by ex vivo luciferase gene assays, SNP rs2736100 was found situated at an intronic enhancer and could pose a genotype-specific impact on *TERT* expression [24].

Since promoter activity is essential for gene expression, the mechanisms between polymorphisms in the *TERT* promoter region and tumorigenesis merit further investigations. A previous functional investigation revealed that the switch with risk allele at *TERT* promoter variant (rs2853669) was pivotal in regulating *TERT* expression by influencing MYC and E2F1 expression. The resultant synergistic effects of MYC/E2F1/TERT expression with the rs2853669 polymorphism further deteriorated the prognosis of prostate cancer [25].

Two synonymous variants at the exon region of *TERT* were identified in this study. SNP rs2736098 was previously found located in an open chromatin region with gene regulatory elements and was reported to correlate with a higher prostate-specific antigen level [26]. The aberrant activation of the related transcriptional factors caused by the variation in this SNP may drive the potential signaling pathways of tumorigenesis [27]. Another synonymous SNP, rs35033501, was reportedly associated with splicing patterns alteration and cancer susceptibility increase by disrupting certain splice sites [28].

Our study identified several novel genetic variants of *TERT* in relation to PCa risk in different ancestries. In European ancestry, five novel SNPs involving intron 12, intron 9, intron 6, and intron 2 regions were discovered, while two novel SNPs situated at intron 2 were discovered in those of Chinese ancestry. This finding can strongly broaden the evidence base on the genetic predisposition of prostate cancer and can refine the genetic architectures of PCa susceptibility loci across the *TERT* region among distinct ancestries. One plausible explanation underlying the association between these novel SNPs and PCa risk may be attributable to a high linkage disequilibrium with adjacent polymorphisms with biological function.

This study supports the evidence of different genetic variants between Europeans and Chinese, which can furnish insights into public health practice, such as different population-based risk stratification and ethnic-specific target screening. According to the findings of our prior research, the precision of risk estimate decreased when applying European-specific PCa polygenic risk score as a risk stratification tool to other ancestry populations [29]. In addition, compared with the large sample size of European GWASs, the small sample size of the East Asian studies led to an imbalance in the risk variants identification and a decrease in the risk estimate precision in this group [30]. Therefore, it is necessary to identify more ethnic-specific risk variants in non-Europeans for a precisely targeted screening and early diagnosis.

There are several limitations in the present study. First, survival data are not available in the Chinese population. However, the Gleason score is associated with PCa survival which may provide some evidence. Second, gene-based analysis is estimated based on the germline loci, whereas the actual function of *TERT* in each individual is unclear. For example, the expression level of *TERT* from each participant is unable to be captured based on the gene-based analysis. Future studies in the biospecimen are considered to address this issue. Finally, subsequent functional experiments are warranted to assess the present findings, as well as to evaluate the different functions between the loci in promoter and intron 2 regions in the *TERT*.

## 5. Conclusions

In conclusion, this study, including a large set of SNPs and two ethnic populations, supports the existing evidence regarding *TERT* polymorphisms in relation to PCa risk and prognosis, discovers some novel PCa-related genetic variants in the *TERT* region, and indicates the heterogeneous genetic architectures of PCa susceptibility loci among distinct ancestries. Further functional studies of *TERT* polymorphisms are required to validate the present findings and reveal the underlying mechanisms.

## Figures and Tables

**Figure 1 cancers-15-02650-f001:**
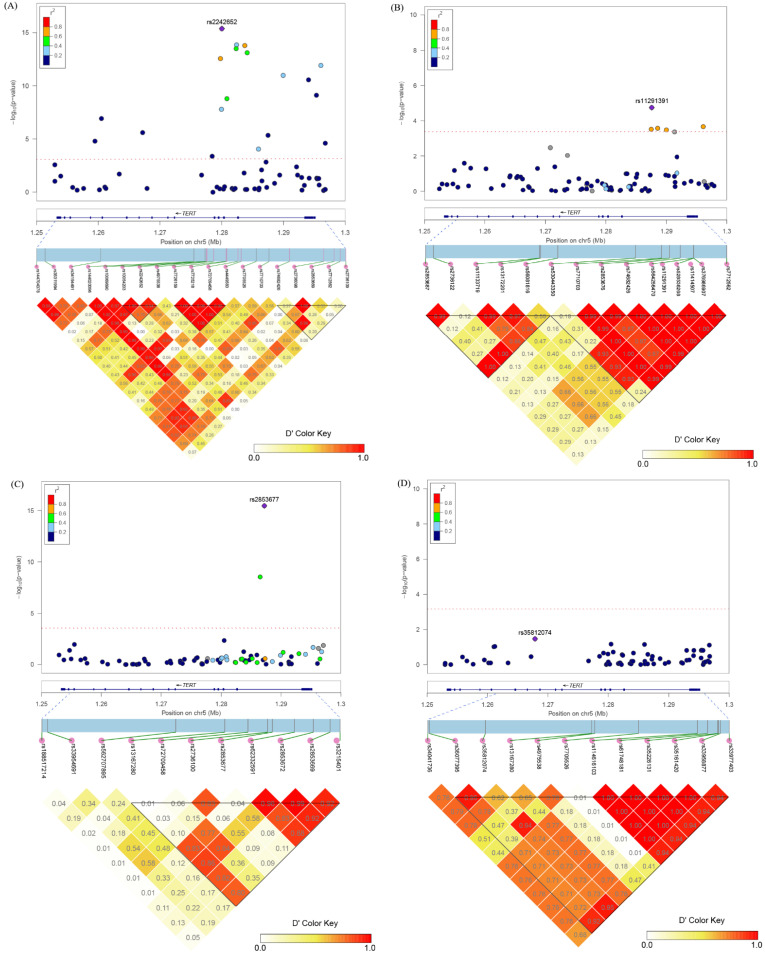
Prostate cancer-related SNPs in the *TERT* locus and their linkage disequilibrium status. (**A**) Prostate cancer-related SNPs in European ancestry; (**B**) Prostate cancer-related SNPs in Chinese ancestry; (**C**) Aggressive prostate cancer-related SNPs in Chinese ancestry; (**D**) Prostate cancer-specific death-related SNPs in European ancestry.

**Figure 2 cancers-15-02650-f002:**
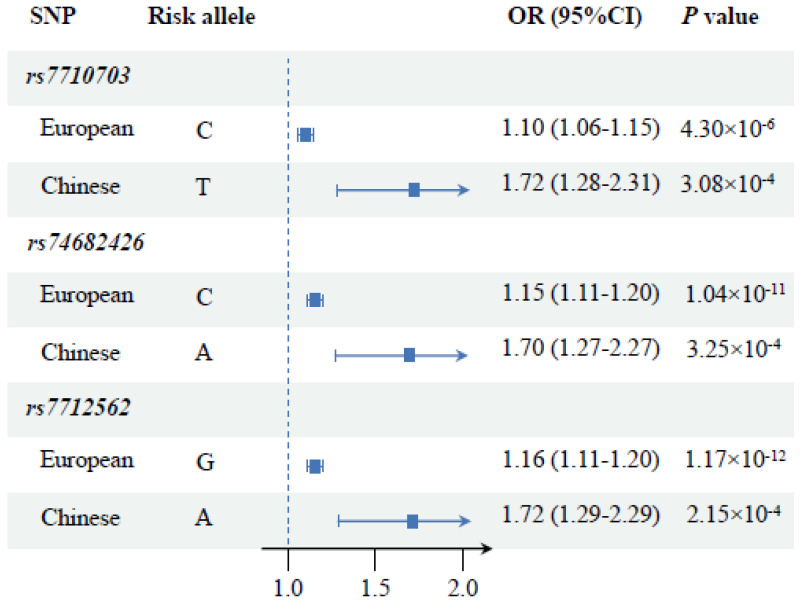
Common susceptibility loci of prostate cancer in European and Chinese ancestry (OR, odds ratio; CI, confidence interval).

**Figure 3 cancers-15-02650-f003:**
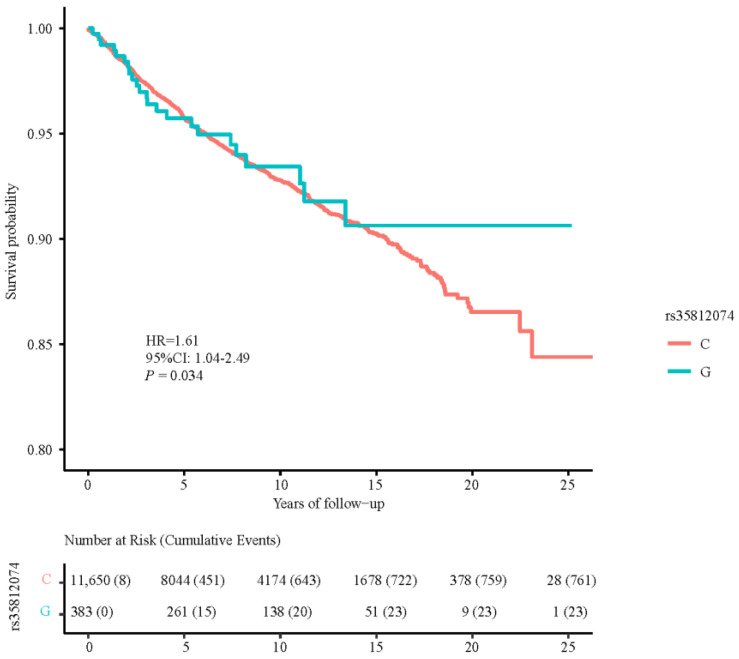
Prostate cancer survival curve of rs35812074 polymorphism. (The survival curve suggested a negative correlation between the C allele of rs35812074 and the survival rate. HR, hazard ratio; CI, confidence interval).

**Table 1 cancers-15-02650-t001:** Characteristics of prostate cancer cases and controls in European and Chinese populations.

	Cases	Controls	*p-*Value
**European**			
N	14,550	195,144	
Age, years (mean ± SD)	62.1 ± 5.5	56.6 ± 8.1	<0.001
Family history	12.4%	8.8%	<0.001
CCI (median, IQR)	2.0 (0.0–3.0)	0.0 (0.0–1.0)	<0.001
**Chinese**			
N	4438	4435	
Age, years (mean ± SD)	70.3 ± 8.0	64.5 ± 9.1	<0.001
Family history	4.2%	3.0%	0.110
PSA value, ng/mL (median, IQR)	21.8 (11.0–71)	9.8 (6.6–14.6)	<0.001
PSA category			<0.001
<1 ng/mL	1.1%	1.6%	
1–4 ng/mL	1.9%	5.4%	
4–10 ng/mL	18.8%	46.6%	
10–20 ng/mL	26.1%	33.6%	
>20 ng/mL	52.1%	12.8%	
Gleason score (median, IQR)	7.0 (7.0–8.0)		
GS category			
2–5	2.1%		
6–7	60.3%		
8–10	37.6%		

SD, standard deviation; IQR, interquartile range; PSA, prostate-specific antigen; PCa, prostate cancer; GS, Gleason score.

**Table 2 cancers-15-02650-t002:** Significant associations between *TERT* SNPs and prostate cancer in European ancestry.

SNP ID	Position *	Location	Alleles ^#^	EAF	OR (95% CI)	*p*-Value
rs144704378 ^†^	1259489	Intron 12	T/C	0.049	1.14 (1.07–1.21)	1.60 × 10^−5^
rs35311994 ^†^	1260514	Intron 12	T/C	0.029	1.22 (1.14–1.32)	1.18 × 10^−7^
rs34194491 ^†^	1267213	Intron 9	C/T	0.025	1.22 (1.12–1.32)	2.50 × 10^−6^
rs144020096 ^†^	1278447	Intron 6	C/A	0.989	1.30 (1.12–1.50)	4.12 × 10^−4^
rs10069690	1279790	Intron 4	C/T	0.743	1.12 (1.09–1.16)	2.58 × 10^−13^
rs10054203	1279964	Intron 4	G/C	0.606	1.08 (1.05–1.11)	1.52 × 10^−8^
rs2242652	1280028	Intron 4	G/A	0.810	1.16 (1.12–1.20)	4.12 × 10^−16^
rs4975538	1280830	Intron 3	G/C	0.648	1.09 (1.06–1.12)	1.57 × 10^−9^
rs7726159	1282319	Intron 3	C/A	0.676	1.12 (1.09–1.15)	3.16 × 10^−14^
rs7725218	1282414	Intron 3	G/A	0.666	1.12 (1.09–1.15)	1.34 × 10^−14^
rs72709458	1283755	Intron 2	C/T	0.799	1.15 (1.11–1.19)	1.57 × 10^−14^
rs4449583	1284135	Intron 2	C/T	0.678	1.12 (1.09–1.15)	7.73 × 10^−14^
rs7705526	1285974	Intron 2	C/A	0.680	1.06 (1.03–1.09)	8.62 × 10^−5^
rs7710703 ^†^	1287505	Intron 2	C/T	0.874	1.10 (1.06–1.15)	4.30 × 10^−6^
rs74682426	1289975	Intron 2	C/A	0.867	1.15 (1.11–1.20)	1.04 × 10^−11^
rs2736098	1294086	Exon 2	T/C	0.280	1.10 (1.07–1.14)	2.60 × 10^−11^
rs2853669	1295349	Promoter	G/A	0.314	1.09 (1.06–1.12)	7.41 × 10^−10^
rs7712562	1296072	Promoter	G/A	0.862	1.16 (1.11–1.20)	1.17 × 10^−12^
rs2736109	1296759	Promoter	T/C	0.407	1.06 (1.03–1.09)	2.52 × 10^−5^

* Locate in Chromosome 5; ^#^ Risk allele/Reference allele; ^†^ Novel susceptibility loci. SNP, single nucleotide polymorphism; EAF, effect allele frequency; OR, odds ratio; CI, confidence interval.

**Table 3 cancers-15-02650-t003:** Significant associations between *TERT* SNPs and prostate cancer in Chinese ancestry.

SNP ID	Position *	Location	Alleles ^#^	EAF	OR (95% CI)	*p-*Value
rs2736100	1286516	Intron 2	A/C	0.540748	1.49 (1.31–1.71) ^†^	2.91 × 10^−9^
rs2853677	1287194	Intron 2	A/G	0.549014	1.74 (1.52–1.98) ^†^	3.52 × 10^−16^
rs7710703 ^‡^	1287505	Intron 2	T/C	0.898	1.72 (1.28–2.31)	3.08 × 10^−4^
rs11291391 ^‡^	1287612	Intron 2	CA/C	0.860	1.73 (1.34–2.25)	3.04 × 10^−5^
rs2853676	1288547	Intron 2	T/C	0.818	1.53 (1.22–1.92)	2.67 × 10^−4^
rs74682426	1289975	Intron 2	A/C	0.893	1.70 (1.27–2.27)	3.25 × 10^−4^
rs7712562	1296072	Promoter	A/G	0.890	1.72 (1.29–2.29)	2.15 × 10^−4^

* Locate in Chromosome 5; ^#^ Risk allele/Reference allele; ^†^ Association with aggressive PCa; ^‡^ Novel susceptibility loci. SNP, single nucleotide polymorphism; EAF, effect allele frequency; OR, odds ratio; CI, confidence interval.

**Table 4 cancers-15-02650-t004:** Gene-based association between *TERT* and prostate cancer in European and Chinese ancestries.

	Gene	Chr	Start	Stop	nSNPs	Z stat	*p-*Value
** *European ancestry* **							
PCa	*TERT*	5	1253282	1295178	56	7.78	3.66 × 10^−15^
PCa mortality	*TERT*	5	1253282	1295178	55	0.95	0.171
** *Chinese ancestry* **							
PCa	*TERT*	5	1253282	1295178	79	1.72	0.043
Aggressive PCa	*TERT*	5	1253282	1295178	54	2.54	0.006

Chr, chromosome; nSNPs, number of single nucleotide polymorphisms included in gene region; PCa, prostate cancer; TERT, Telomerase reverse transcriptase.

## Data Availability

UKB Data used in this research are publicly available to qualified researchers on application to the UK Biobank (www.ukbiobank.ac.uk (accessed on 15 November 2022)). The study protocol, statistical analysis plan, and analytical code of this study will be available from the time of publication in response to any reasonable request to the corresponding author.

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
