# Peer review of "Genetic Polymorphisms of the Telomerase Reverse Transcriptase Gene in Relation to Prostate Tumorigenesis, Aggressiveness and Mortality: A Cross-Ancestry Analysis"

_cancers, 2023, doi:10.3390/cancers15092650_

Round 1

Reviewer 1 Report

Since this is a study of the risk of developing prostate cancer for SNPs in TERT in a dry analysis using a large database, I am afraid that there is not much to go into.

Major 

1. the study on the risk of developing prostate cancer by TERT itself is not novel.

2. In the Discussion section, I think the treatment has little applicability based on the present results.  In the Discussion section, the possibility of applying the results of this study to treatment should be discussed based on the differences in SNPs between Europeans and Chinese. 

3. the number at risk in Fig. 3 should be mentioned

4. The outcome is defined as cancer death or aggressiveness (defined by GS/PSA), but I think there is too little information on the patient's background (Table 1). Please some more detailed information. 

Translated with www.DeepL.com/Translator (free version)

Author Response

Dear reviewer,

We would like to thank you for your careful and thorough consideration of our manuscript, and your recognition of this work. Your comments are all valuable and very helpful for revising and improving the manuscript, as well as the important guiding significance to our research. We have read all the comments carefully and have made corrections that we hope to meet with approval.  

[1] Reviewer’s comment: The study on the risk of developing prostate cancer by TERT itself is not novel.

【Our Response】We agreed that the association between TERT and PCa has been reported by prior research; nevertheless, several remarkable novelties should be noted in this study. First, this is the first cross-ancestry study using individual data on this topic. Second, the current study contains the largest association study in Chinese population, and several PCa susceptibility loci in TERT were newly identified. Third prior research only focused on one outcome, mostly diagnostic PCa, but our study comprehensively illustrates the association of TERT polymorphisms with prostate tumorigenesis, PCa aggressiveness, and PCa-specific death.

[2] Reviewer’s comment: In the Discussion section, I think the treatment has little applicability based on the present results. In the Discussion section, the possibility of applying the results of this study to treatment should be discussed based on the differences in SNPs between Europeans and Chinese.

【Our Response】Many thanks for your suggestion. Although the clinical treatment has little applicability, the present results may have more public health implications such as population-based risk stratification and target screening. According to the findings of our previous study (Ruan X, et al. Application of European-specific polygenic risk scores for predicting prostate cancer risk in different ancestry populations. Prostate. 2023,83(1):30-38. DOI: 10.1002/pros.24431), the precision of risk estimate decreases when we apply European-specific PCa polygenic risk score as a risk stratification tool to other ancestry population. Thus, it’s necessary to identify the ethnic-specific risk variants in non-Europeans for a precisely targeted screening and early diagnosis, as indicated by a recent review (Conti DV, et al. rans-ancestry genome-wide association meta-analysis of prostate cancer identifies new susceptibility loci and informs genetic risk prediction. Nat Genet. 2021,53(1):65-75. DOI: 10.1038/s41588-020-00748-0). We have added a paragraph in the discussion section based on this point.

[3] Reviewer’s comment: The number at risk in Fig. 3 should be mentioned.

【Our Response】We have added it to the Figure.

[4] Reviewer’s comment: The outcome is defined as cancer death or aggressiveness (defined by GS/PSA), but I think there is too little information on the patient's background (Table 1). Please some more detailed information.

【Our Response】We have added the PSA category and GS category information to Table 1.

Best regards.

Reviewer 2 Report

The number of controls for the European patient cohort are significantly high compared to the PCa patient cohort. The numbers are not comparable as they are with the Chinese patient cohort numbers that are closer. The authors state at the end of the abstract that there is no association with the TERT expression in both patient groups but contradicts themselves in the last sentence, please clarify this discrepancy. 

Line 42: change "In US" to "In the US"

Line 43: change "death" to "deaths"

Line 143: remove the word "two" 

Author Response

Dear reviewer,

We would like to thank you for your careful and thorough consideration of our manuscript, and your recognition of this work. Your comments are all valuable and very helpful for revising and improving the manuscript, as well as the important guiding significance to our research. We have read all the comments carefully and have made corrections that we hope to meet with approval.  

[1] Reviewer’s comment: The number of controls for the European patient cohort are significantly high compared to the PCa patient cohort. The numbers are not comparable as they are with the Chinese patient cohort numbers that are closer.

【Our Response】Thanks for your query. This is due to the difference of study design in two populations: general population-based cohort study design in the Europeans and case-control design in the Chinese. As all the analyses in this research were conducted separately in two populations, and the interpretation of the results were based on the two different populations, we believed that such limitation would not change our conclusion.

[2] Reviewer’s comment: The authors state at the end of the abstract that there is no association with the TERT expression in both patient groups but contradicts themselves in the last sentence, please clarify this discrepancy.

【Our Response】Sorry for the unclear statement, but we did not mention “TERT expression” in the abstract. The finding of null association is between TERT gene and PCa death at a gene-based level analysis. This may be due to the limited susceptibility loci included in this analysis at the current stage. Whereas in the SNP association analysis, one TERT SNP (rs35812074) is found related to PCa death. This study is also the first to identify a risk locus in TERT that can affect PCa-specific death. We believe that with larger sample sizes available, more risk variants relevant to PCa death could be identified in the future. We have changed the statement to be more clear.

[3] Reviewer’s comment: Line 42: change "In US" to "In the US"; Line 43: change "death" to "deaths"; Line 143: remove the word "two"

【Our Response】Thanks for your advice. We have changed correspondingly.

Yours sincerely.

Reviewer 3 Report

In the article “Genetic Polymorphisms of the Telomerase Reverse Transcriptase Gene in Relation to Prostate Tumorigenesis, Aggressiveness and Mortality: A Cross-ancestry Analysis”. The authors found TERT polymorphisms were found to be associated with prostate cancer (PCa) risk, aggressiveness, and cancer death. Nineteen susceptibility loci, including 5 novel ones, were detected in Europeans, while 7 loci, including 2 novel ones, were discovered in Chinese. SNPs rs2736100 and rs2853677 were significantly associated with aggressive PCa, while rs35812074 was marginally related to PCa death. Gene-based analysis showed a significant association of TERT with PCa and PCa severity but not with PCa death.

This research is meaningful, but there are some questions that the authors must address.

1.       In "2.4, SNPs imputation and quality control", the authors need to specify the phasing tool used prior to imputation.

2.       The rationale of using different imputation algorithms on different population data also needs to be specified.

3.       For Figure 3, could you please provide the numbers of case survival for every 5 years? Since the authors were analyzing 2 large cohorts, it seems strange that no one died during the 12-25 years of follow-up.

4.       Continuing from the previous question. The minor allele frequency of rs35812074 is quite low in gnomAD dataset. This may different between European and Chinese. What is the MAF in Chinese population data?

Author Response

Dear reviewer,

We would like to thank you for your careful and thorough consideration of our manuscript, and your recognition of this work. Your comments are all valuable and very helpful for revising and improving the manuscript, as well as the important guiding significance to our research. We have read all the comments carefully and have made corrections that we hope to meet with approval.  

[1] Reviewer’s comment: In "2.4, SNPs imputation and quality control", the authors need to specify the phasing tool used prior to imputation.

【Our Response】Thanks for your suggestion. The phasing tools are “SHAPEIT” for Europeans and “Eagle v2.4” for Chinese. We have added it to the Method section.

[2] Reviewer’s comment: The rationale of using different imputation algorithms on different population data also needs to be specified.

【Our Response】Thanks for your insightful comments. The imputation algorithms are “IMPUTE4” for Europeans and “minimac4” for Chinese. According to a prior publication, the imputation accuracy of these two algorithms was highly consistent even using different reference panels (e.g., 1000G Phase 1: r2=0.77 vs. r2=0.77; 1000G Phase 3: r2=0.79 vs. r2=0.79; HGC v1.1: r2=0.90 vs. r2=0.90) (Das S,et al. Next-generation genotype imputation service and methods. Nat Genet. 2016,48(10):1284-1287. DOI: 10.1038/ng.3656). We have specified this rationale in the Method section.

[3] Reviewer’s comment: For Figure 3, could you please provide the numbers of case survival for every 5 years? Since the authors were analyzing 2 large cohorts, it seems strange that no one died during the 12-25 years of follow-up.

【Our Response】Thanks for your query. We have added the Table “number at risk (events)” into the Figure 3. Since the MAF of rs35812074 is very low (MAF=0.016) in the UKB cohort, the number at death risk during the 12-25 years are quite limited (n~50) among patients carrying the G allele of rs35812074. This may result in the null death cases as shown in the modified figure.

[4] Reviewer’s comment: Continuing from the previous question. The minor allele frequency of rs35812074 is quite low in gnomAD dataset. This may different between European and Chinese. What is the MAF in Chinese population data?

【Our Response】 Thanks for your query. The MAF of rs35812074 is 0.042 in ChinaPCa data. In gnomAD dataset, the minor allele frequency of this SNP is 0.064 in East Asian population.

Yours sincerely.